# Effects of Personality Traits on the Food-Scratching Behaviour and Food Intake of Japanese Quail (*Coturnix japonica*)

**DOI:** 10.3390/ani11123423

**Published:** 2021-12-01

**Authors:** Xinyu Zhang, Xue Wang, Wei Wang, Renxin Xu, Chunlin Li, Feng Zhang

**Affiliations:** 1School of Resources and Environmental Engineering, Anhui University, Hefei 230601, China; zxy199609@126.com (X.Z.); XueWang1108@163.com (X.W.); 18846056229@163.com (R.X.); fzhang188@163.com (F.Z.); 2Gangcha Forestry and Grassland Bureau, Qinghai 812300, China; 18815615165@163.com; 3Anhui Province Key Laboratory of Wetland Ecosystem Protection and Restoration, Anhui University, Hefei 230601, China

**Keywords:** foraging behaviour, animal personality, food-scratching behaviour, Japanese quail

## Abstract

**Simple Summary:**

Scratching can help animals find more buried food and thus is an important food-searching behaviour for ground-feeding birds such as gamebirds. Due to the existence of animal personality, individuals within a population may exhibit different food-scratching patterns. This study tested the impacts of personality traits (i.e., boldness and exploration) on food-scratching behaviour and food intake of Japanese quail (*Coturnix japonica*). We found that boldness and exploration were repeatable, respectively, and were correlated. When entering a food patch, proactive (i.e., bolder and more explorative) quails scratched for food earlier and more frequently with a longer time. Frequent and longer food-scratching may motivate longer foraging time in proactive quails which can get more food intake. The correlation between personality and food intake was sex dependent. Proactive females had more food intake during the first half of the foraging process and the correlation became weak as time went on. The pattern was opposite in males. In conclusion, our study suggests that personality traits have significant effects on animals’ food-searching strategies which may be correlated with their foraging success and fitness.

**Abstract:**

Overall foraging success and ultimate fitness of an individual animal is highly dependent on their food-searching strategies, which are the focus of foraging theory. Considering the consistent inter-individual behavioural differences, personality may have a fundamental impact on animal food-scratching behaviour, which remains largely unknown. In this study, we aimed to investigate how personality traits (i.e., boldness and exploration) affect the food-scratching behaviour and food intake of the domestic Japanese quail *Coturnix japonica* during the foraging process. The quails exhibited significant repeatability in boldness and exploration, which also constituted a behavioural syndrome. More proactive, that is, bolder and more explorative, individuals scratched the ground more frequently for food and began scratching earlier in a patch. Individuals that scratched more frequently had a longer foraging time and a higher food intake. The correlation between personality traits and temporary food intake during every 2 min varied over time and was sex dependent, with females exhibiting a positive correlation during the first half of the foraging stage and males after the initial stage. These findings suggest that personality traits affect the food-scratching behaviour and, thus, the food intake of quails. Our study provides insights into the impact of personality traits on animal’s foraging behaviour by influencing their food-searching strategies.

## 1. Introduction

When individual forage optimally, they can increase their success at collecting food and minimising energetic costs [1]. Survival is therefore dependent on performing successful foraging strategies. Exploring the factors influencing foraging behaviour may help to understand how animals adapt to the environment and improve their fitness [2]. Previous studies on foraging behaviour mainly focused on the choice of patch type, the time budgets in patches, and how to move between patches [3,4]. Comparatively, little is known about how animal personality influences their food-searching strategies and, thus, their foraging success in a food patch.

Searching strategy within a food patch is of great significance for ground-feeding birds that often use their claws to scratch grounds for buried food [5,6]. The scratching behaviour is correlated with food acquisition, and the mode, posture, and frequency of scratching behaviour may vary among and within species, with important implications for animal foraging success [7]. For example, the scratching time of white-throated sparrows *Zonotrichia albicollis* is usually longer than that of dark-eyed juncos *Junco hyemalis* [7]. This difference could be explained by the differences in the microhabitats between the two species, with white-throated sparrows foraging in heavily littered areas. Rufous-collared sparrows *Zonotrichia capensis* which often scratch a patch can get more seeds buried in the litter, while conspecifics that do not use this searching method can only gain unburied seeds [8]. The inter-individual differences in using scratching behaviour may be attributed to their differences in energy requirements, and strong scratching ability may help find more food for individuals with higher energy needs [9,10].

Food-searching strategies may vary largely among individuals because of the existence of animal personality, which quantifies consistent inter-individual behavioural differences [11]. Animal personality has been widely found in both captive and wild animal populations, with implications for their fitness [9,12]. As proposed by Carere et al. [13], researchers often measure one or more of five personality traits, i.e., shyness–boldness, exploration, aggressiveness, activity, and sociability. Behaviours are often correlated across contexts, constituting behaviour syndromes, allowing characterising individuals on main axes summarising the correlated behaviours [14]. For example, boldness and exploration are often correlated and thus individuals can be characterised on the proactive–reactive axis [15]. Proactive individuals (i.e., bolder and more explorative) often exhibit a high propensity to take risks, and fast and superficial exploration, whereas reactive individuals are characterised by a low propensity to take risks and slow but thorough exploration [15]. The existing literature focuses primarily on the effects of personality traits on pre-foraging decision-making, for example, which resources to use, where to forage, and whether to forage independently or in groups [16,17]. Further studies are needed to investigate the influence of personality traits on food-searching methods, such as the scratching behaviour of ground-feeding birds of the Order Galliformes (the pheasants, partridge, grouse, and allies), and thus food gains within food patches.

In this study, we investigated whether personality traits affect food-scratching behaviour and food intake in Japanese quails (*Coturnix japonica*) during foraging. We first quantified each quail’s personality traits, including boldness and exploration, and then tested the effects of personality traits on the pattern of quails’ scratching behaviour and food intake. Studies on birds and mammals suggest that proactive individuals usually adopt active foraging strategies [8]. Therefore, we expected that proactive quails would scratch the ground earlier when reaching a patch and scratch more frequently to find food during foraging. More frequent scratching may use more energy and thus proactive individuals need to find more food to maintain their energy needs [9]. Given that personality traits are sex dependent in many species [18], we further predicted that the effect of personality on scratching behaviour may vary between males and females.

## 2. Materials and Methods

### 2.1. Ethical Note

The experiments complied with the current animal welfare and scientific research ethics legislation in China. All animal care and experimental procedures were approved by the Institutional Animal Care and Use Committee of Anhui University (permission no. 2020-037), and the quails were not harmed during the experiments. After the study was completed, the quails were kept in the cardboard containers in the laboratory for other behavioural studies.

### 2.2. Study Species and Breeding Conditions

Batches of more than 2000 mixed-sex Japanese quails were incubated for 16 d at a quail farm in Furong District, Changsha city, Hunan Province, China. The incubation temperature was 37 °C during the first day of incubation, with a daily decline of 0.5 °C until room temperature (25 °C). The newly hatched quails were raised according to the standard rules of care and space for quails [19]. Specifically, they were exposed to a natural photoperiod (~14:10 h, light:dark photoperiod) after incubation and were fed twice a day (9 a.m. and 6 p.m.). The basal diet contained 57% corn, 30% soybean meal, 5% crude protein, 5% fish meal, 2% stone meal, and 1% soybean oil. Water was provided ad libitum, and daily maintenance was performed at 8 a.m.

Upon sexual maturity (approximately 60 d) of the quails in each batch, approximately six healthy quails (three females and three males) were randomly selected and transferred to the laboratory at the School of Resources and Environmental Engineering at Anhui University. We bought quails from the farm in 12 successive batches within two months (from August to October 2020); resulting in a total of 72 subjects (38 females and 34 males) at approximately the same age when being tested. On arrival, the experimenter determined the sex of the quail based on its feather colour and anus. The quails were then individually housed in labelled opaque cardboard containers (50 cm long, 40 cm wide, 60 cm high; hereafter, housing container), each with a white ceramic tray (40 cm long, 35 cm wide, 3 cm high). The ceramic trays were sprinkled with sand for quails to scratch and enjoy sand bathing [20]. Two Petri dishes (90 mm diameter, 15 mm depth) were placed on a ceramic tray, one containing food and the other containing water. The top of the container was covered by a white net to prevent the quail from escaping, while allowing cleaning of the ceramic tray and replacing the Petri dish. Daily cleaning was carried out at 8 a.m., and twice-daily feeding chores were given at 9 a.m. and 6 p.m. The quails were exposed to a natural photoperiod, and the room temperature was maintained at 25 °C.

In the 3 d prior to the experiments, the quails were habituated to the food used in the trials, that is, small yellow millets (MILLET, Wuchang Rice Products Co., Ltd., Wuhan, China; 18% energy, 17% protein, 6% lipid, 24% carbohydrates), which were sieved to a uniform size (1 mm diameter and weighs approximately 0.002 g). The quails were not fed 24 h before the experiments, so hunger motivated them to search for food during the trials.

### 2.3. General Experiment Procedure

We first used open arena assays to measure boldness and exploration individually for all focal subjects (see the subsection of Personality assays for details) to test their behavioural repeatability and syndrome. Subsequently, we conducted foraging behaviour trials for all quails (see the subsection of Foraging behaviour experiments for details) to quantify their scratching patterns and food intake. Personality trials for each focal individual were repeated three times each on successive days. Foraging trials for each focal individual were repeated twice, following the personality trials each day. The trial sequences were randomised, and all experiments were carried out in the same laboratory with sufficient light and constant temperature (25 °C). To avoid potential interference from the experimenters, the trial devices were surrounded by a 1.5 m high opaque curtain. To control for the effect of body weight on scratching behaviour and food intake of quails, each subject was weighed (0.1 g) at the end of the experiments for a batch of quails.

### 2.4. Personality Assays

We measured boldness and exploration continuously in an open arena (195 cm long, 165 cm wide, 150 cm high), which was a rectangular field surrounded by gray opaque curtains (Figure 1). A camera (Sony HDR-CX510, 55× extended zoom, Sony Corporation, Tokyo, Japan) was placed above the arena to record the behaviour of the quail. An opaque cardboard box (the same as the housing container; hereafter, referred to as the initial refuge) was fixed outside the middle of one end of the arena. There was a sliding trapdoor (15 cm × 15 cm) on the side of the initial refuge facing the arena. The experimenter could remotely pull the fishing line tied to the trapdoor to allow the quail to walk from the initial refuge to the arena without interference. The ground of the field was divided into 143 squares (15 cm × 15 cm), marked by dark lines. There was a Petri dish (same as that in the housing container) with 20 artificial plant leaves (red and green; 3 cm long, 2 cm wide), 30 cm in front of the trapdoor. The rest of the field was equipped with nine novel objects, not allowing quails to thoroughly view the arena at an immediate glance. To avoid potential habituation to the objects, we used novel materials with the same shape (triangular Decca; 20 cm long, 10 cm high) but different colours that is, white, red, and blue, in the three repeated trials. Materials of different colours were scattered in the same configuration for different subjects. As boldness measures individuals’ willingness to enter an environment with potential predation risk [21,22], we simulated the risk of predation presented to the quails using a hawk model (26 cm wingspan, 6 cm high, and weighs approximately 50 g), a common predator in the wild. The model was suspended by a fishing line, 90 cm in front of the trapdoor, and the experimenter pulled the model from the ground to a height of 1.5 m.

At the beginning of the trial for boldness and exploration, a randomly selected quail was gently transferred from the housing container to the initial refuge, and then the camera began recording. The subject was allowed to acclimate to the initial refuge for 5 min. Next, the trapdoor was remotely opened and remained open until the end of the experiment. Meanwhile, the experimenter remotely pulled the hawk model flying at a constant speed (approximately 10 cm/s) from the ground to a height of 10 cm. The hawk model was suspended at 10 cm above the ground until the quail came out of the refuge, after which the model was slowly pulled away from over the experimental field using the fishing line. The subject was given a maximum of 20 min to emerge from the initial refuge, and boldness of the subject was defined as 20 min minus the time taken to emerge [23], with bolder individuals emerging earlier from the shelter. We considered a quail to have come out from the initial refuge when its whole body crossed the trapdoor. The camera continued to record the movement of the subject for 12 min after it left the initial refuge. If the quail did not leave the refuge within 20 min, we terminated the experiment and determined its boldness score as 0 s. In this case, the experimenter gently moved the quail from the refuge to the arena, and the camera continued to record for 12 min. We defined the last 10 min for each subject in the arena as its exploration trial. We extracted 600 image stacks from the 10 min movement videos (one frame per second) and used Image J (http://rsbweb.nih.gov/ij/; accessed on 7 October 2019) to monitor the movement of the quail. Following Bergvall et al. [24], we used the total number of squares that the quail passed without repetitions to quantify its exploration score.

### 2.5. Foraging Behaviour Experiments

We carried out the foraging behaviour experiments in an arena that was like the housing container (50 cm long, 40 cm wide, 60 cm high). A camera was fixed above the arena to record the foraging behaviour of the quail during the entire process of each experiment. A Petri dish (90 mm diameter, 15 mm depth) was placed on a white ceramic tray (40 cm long, 35 cm wide, 3 cm high) that was placed on the floor of the cardboard container to collect food that might be thrown outside the Petri dish. The top of the arena was covered with a white net to prevent the quail from escaping, while allowing the camera to record the subject during the experiment.

At the beginning of the foraging experiment, the camera was turned on, and a quail was transferred from the housing container to the experimental arena. We mixed 10 g small yellow millet sieved to the same size (1 mm diameter and weighed approximately 0.002 g) with 20 artificial plant leaves (same as those in the personality assays) in the Petri dish to simulate the buried food resources in the wild [25]. The experimenter gently placed the Petri dish back into the arena and monitored the subject from a distance through the camera screen. The experiment started when the quail began to feed and the subject’s feeding was interrupted every 2 min by the experimenter, taking out the ceramic tray and the Petri dish to weigh the remaining millet (accurate to 0.001 g). The remaining millet was again mixed with the leaves, and then returned to the arena. The measurement interruption lasted for approximately 1 min. The following 2 min foraging periods also started when the subject resumed feeding and were ended by the experimenter, taking out the Petri dish to weigh the remaining millet. The Petri dish and the ceramic tray were placed in the same location for different individuals. During the pilot experiments, we found that most quails stopped foraging within 15–20 min, and almost all quails stopped feeding within 24 min. Therefore, we determined the total length of the foraging experiment as 24 min, during which there were 12 phases of 2 min foraging, and food intake during each 2 min foraging period was referred to as periodic food intake. At the end of the experiment, the quail was transferred back to its housing container, and the ceramic tray was cleaned to remove chemical traces from the previous tests. From the videos, we quantified the following four variables related to food scratching behaviour for each subject over the whole foraging experiment (i.e., containing all the 12 2 min stages): (1) the total number of scratching bouts during which the quail continuously scrabbled the Petri dish with its claws (NSB), (2) the number of scrabbles during each scratching bout (NS), (3) the time of the first scratch since the beginning of experiment (TFS), and (4) the total time of scratching bouts (TTS). Scratching behaviour was defined as a quail using its claws to scrabble the artificial plant leaves for food.

### 2.6. Statistical Analyses

A generalised linear mixed-effects model was fitted for each sex to measure the repeatability of boldness and exploration and to partition their phenotypic correlation into among and within-individual (residual) components. The two log_10_-transformed behaviours were concurrently included in each model as response variables, while individual ID was included as a random effect. Following Dingemanse et al. [26], the model was fitted using the R package *MCMCglmm* to test the behavioural repeatability and syndrome (among-individual correlation). These models were run for 220,000 iterations after a burn-in of 20,000 iterations and were thinned by 25 iterations.

The boldness and exploration scores for each quail were averaged from the three trials, and the four variables related to scratching behaviour were averaged from the two foraging trials. These means were found to be normally distributed (*p* > 0.05; the Shapiro–Wilk Test) and thus were directly used in the following analyses. Given that the two personality traits were strongly correlated (see the subsection of Behavioural repeatability and correlation in Results for details), they could not be used together as explanatory variables in linear models because of the lack of independence when estimating their partial effects. Therefore, we used the R package psych to conduct a principal component analysis (PCA) to derive two new variables, the principal components 1 and 2 (PC1 and PC2; orthogonally rotated), maximising the variance of these variables among individual quails [27]. PC1 (loadings: 0.707 boldness + 0.707 exploration; eigenvalue: 1.296) explained 84.0% of the total variance, while PC2 explained 16.0% (loadings: −0.707 exploration + 0.707 boldness; eigenvalue: 0.565). Quails that had high scores on PC1 took less time to emerge from the initial refuge and explored a larger area, so it can be considered as a measure of proactive which includes both boldness and exploration. Quails that had high scores on PC2 took less time to emerge from the shelter but explored a small area. To test whether personality traits have effects on scratching behaviour, we used general linear models (GLMs) to fit the effects of PC1, PC2, body weight, and sex on the four scratching behaviour variables. The two-way interactions between body weight, sex, and PC1 and PC2 were initially included, but were excluded from the final models because of no significant effects (*p* > 0.05).

Considering “S-shape” increasing curve, we used the following logistic function to fit the accumulative food intake (denoted by *y*) over time (i.e., a function with regard to time) for each individual:(1)$$y=\frac{se^{rt}}{k+e^{rt}}$$
where *s* indicates the asymptotic food intake (i.e., maximum value), *k* reflects the half-saturation status, *r* is a parameter describing the speed of food intake, and *t* is time. The goodness of fit of the logistic model for all individuals was over 96%. We determined that the quail was full when *y* reached *s* × 95% (saturated appetite) and defined this time as the saturation time. We used GLMs to fit the effects of PC1, PC2, body weight, and sex on saturated appetite and saturation time, respectively. The two-way interactions between body weight, sex, and PC1 and PC2 were also excluded from the final models because of the lack of significant effects (*p* > 0.05). To test the change of the relationship between personality traits and periodic food intake along foraging time and its difference between sexes, the Pearson’s correlation coefficient between personality traits and periodic food intake during each 2 min foraging period was calculated and tested using GLM for the effects of sex, foraging time, and their two-way interaction.

All statistical analyses were performed with R 3.6.3 [28]. The data are shown as mean ± standard error (SE) and the significance level was *p* < 0.05.

## 3. Results

### 3.1. Sampling Information

A total of 72 individuals (38 females and 34 males) were tested in the personality and foraging experiments. The average weight of the tested quails was 100.1 ± 6.8 g in females and 113.3 ± 10.0 g in males.

### 3.2. Behavioural Repeatability and Correlation

Both females and males showed significant repeatability in boldness (females: 0.363 < *r* < 0.715; males: 0.556 < *r* < 0.837; *r* denotes the posterior mean of repeatability) and exploration (females: 0.505 < *r* < 0.798; males: 0.252 < *r* < 0.672) across assays. At the phenotypic level, both females (95% confidence interval: 0.615 < *r* < 0.832; *r* denotes the correlation coefficients) and males (0.281 < *r* < 0.656) exhibited significant positive correlations between boldness and exploration. At the among-individual level, the two behaviours were also positively correlated, indicating a behavioural syndrome in both females (0.905 < *r* < 1.000) and males (0.571 < *r* < 0.999). At the within-individual level, the positive correlation was only significant in females (0.154 < *r* < 0.534), but not in males (−0.230 < *r* < 0.275).

### 3.3. Personality Traits and Food-Scratching Behaviours

The total number of scratching bouts (NSB; 9.764 ± 12.429) and the number of scrabbles during each bout (NS; 29.681 ± 46.680) were positively correlated with PC1. The time of the first scratch since the beginning of experiment (TFS; 401.194 ± 535.577 s) was negatively correlated with PC1 and PC2. The total time of the scratching bouts (TTS; 15.465 ± 24.294 s) was positively correlated with PC1. The proactive quails showed higher levels of scratching behaviours, which did not differ between sexes and were not affected by body weight. These results indicated that proactive, that is, bolder and more explorative, individuals scratched the ground earlier and more frequently for food in a patch (Table 1).

### 3.4. Personality Traits and Food Intake

The gain curves of proactive quails were generally above the reactive individuals in the same coordinate system, which indicated that proactive individuals ate more food than reactive individuals (Figure 2). The saturation time (*t* = 2.394, *p* = 0.020; Figure 3a) and saturated appetite (*t* = 4.937, *p* < 0.001; Figure 3b) were positively correlated with PC1, indicating that, compared with reactive individuals, proactive ones would be saturated after a longer time and consuming a greater amount of food. Females had a longer saturation time (*t* = −1.787, *p* = 0.079) and a larger saturated appetite (*t* = −1.756, *p* = 0.084) than males. The correlation between PC1 and periodic food intake varied over time (*t* = −7.577, *p* < 0.001) and was sex dependent (*t* = −7.109, *p* < 0.001). The correlation was significant during early foraging, but non-significant during later periods in females. The pattern was the opposite in males (Table 2).

## 4. Discussion

Consistent with many studies in other species [29,30], we found repeatability in boldness and exploration, which were positively correlated at both the phenotypic and among-individual levels, with the latter indicating a behaviour syndrome in domestic Japanese quails. The widely found behavioural consistency and correlation have profound ecological and evolutionary implications, which may affect the ways in which an individual interacts with environments and, thus, its fitness [31]. Boldness refers to the willingness to take risks; therefore, bolder individuals might explore more in a novel environment that may be dangerous [32]. It is well established that boldness and exploration have fitness consequences and influence important life-history decisions such as foraging strategies [33]. For example, bolder individuals feed earlier after a period of food deprivation [34], and more explorative individuals usually lead food-searching in flocks [35]. In this study, we found that boldness and exploration have impacts on food-scratching behaviour and food intake in Japanese quails, which are discussed below.

We found that proactive quails scratched a food patch more frequently and spent more time scratching than reactive birds (Table 1). For ground-feeding birds, especially Galliformes, scratching with claws is an important and efficient food-searching strategy to increase the probability of food acquisition [36]. However, frequent food-scratching behaviour may expose animals to predators because the scratching behaviour distracts the forager’s attention on potential risks and predators can easily find active prey with loud noise [37,38]. Therefore, proactive individuals are expected to scratch more in patches than reactive individuals. Although they may have a higher risk of being preyed upon, proactive individuals can get more food through more scratches. On the contrary, reactive individuals may scratch less to reduce predation risks at the expense of food acquisition. In addition to scratch frequency and time spent scratching, we also found that more proactive individuals would start food-scratching earlier when they first enter the food patch, which may reflect the animals’ responses to new environments. Bold individuals often initiate earlier exploration for buried food in new environments. Earlier and more frequent scratches may help proactive individuals find more food of higher quality, while reactive individuals are more likely to eat unburied food first and not consider the quality of the food because they are more intent on assessing the risk of a patch [8]. These findings suggest that various personality types may facilitate different food-searching strategies to coexist within populations to adequately find and consume food resources in their habitats.

Proactive quails foraged for a longer time during the whole 24 min foraging experiment (Figure 3), which may be explained by a higher motivation to forage. Frequent scratching may help proactive individuals find more food which may facilitate motivation to forage and thus increase feeding time [39]. Proactive individuals have greater maximum metabolic rates than reactive individuals [40]. Higher foraging motivation is expected to help proactive individuals acquire more food to maintain their higher metabolic levels related to frequent food-scratching behaviours. This finding is consistent with those of previous studies. For example, compared with shy conspecifics, bold Namibian rock agama *Agama planiceps* eat more because they have longer foraging time and larger home ranges [41]. In addition to the effect of personality, foraging time and food intake may differ between sexes [42]. For example, in a series of foraging tasks, female red-chested tamarin *Saguinus labiatus* have a longer foraging time and ate more food than males [43]. We also found that female quails had longer foraging time and larger saturated appetites. As in other species, female quails have higher energy and nutrient demands which need longer foraging time [44].

Boldness and exploration were positively correlated with periodic food intake throughout the foraging experiments (Table 2), suggesting that proactive individuals consistently had a higher food intake. This is consistent with our finding that proactive individuals show more frequent food scratching than reactive ones, if food scratching leads to increased food intake, as discussed above. Interestingly, the influence of personality traits on periodic food intake was not consistent throughout the foraging experiment and differed between the sexes. Specifically, female personality traits had a prominent effect on food intake during early foraging, and this effect gradually decreased and diminished when their appetites were gradually satisfied. The pattern of the effect was the opposite in males (Table 2). The underlying reason might be related to sex differences in the trade-offs between foraging and other activities [45]. When reaching a food patch, the main motivation of females is to obtain food [46], while males also exhibit territorial defence behaviours [47]. Therefore, the effect of personality traits on food intake during earlier foraging periods can be detected in females but may be masked by other activities in males. Along with foraging, the female’s appetite was gradually satisfied; thus, the influence of personality traits on food intake decreased. On the contrary, after the initial expression of defending behaviours, males focused on feeding; thus, the effect of personality traits became prominent.

Our study provides further evidence for personality in animals and its effect on animals’ foraging behaviour and food intake. The foraging experiments were carried out in a single food patch and separately for each quail. However, animals may encounter more patches while searching for food and they often move between these patches. How personality affects animals’ use of and movement between these food patches remains unstudied, which needs more research to provide further insights. Furthermore, gregarious animals often forage in groups. Therefore, further studies can pay attention to the effects of personality traits on foraging behaviours in groups and their interactions with animals’ social behaviours.

## 5. Conclusions

We found that foraging behaviour in domestic Japanese quails was influenced by personality traits, that is, boldness and exploration, which showed consistent inter-individual differences and comprised a behavioural syndrome. Proactive (i.e., bolder and more explorative) individuals used food-scratching behaviour more frequently and foraged for longer to obtain more food. When reaching a food patch, proactive individuals also exhibited food-scratching behaviour earlier, indicating that proactive individuals have faster responses to new environments. Proactive individuals had longer foraging time and larger saturated appetites to meet their energy needs. Boldness and exploration were positively correlated with periodic food intake throughout the foraging process, resulting in a larger total food intake in proactive individuals. Personality traits and periodic food intake of females were positively correlated during early foraging in a food patch, but the correlation became weak as time went on. The pattern of correlation was the opposite in males. These findings suggest that personality traits may have a significant influence on animals’ foraging success and fitness by affecting searching strategies in food patches.

## Figures and Tables

**Figure 1 animals-11-03423-f001:**
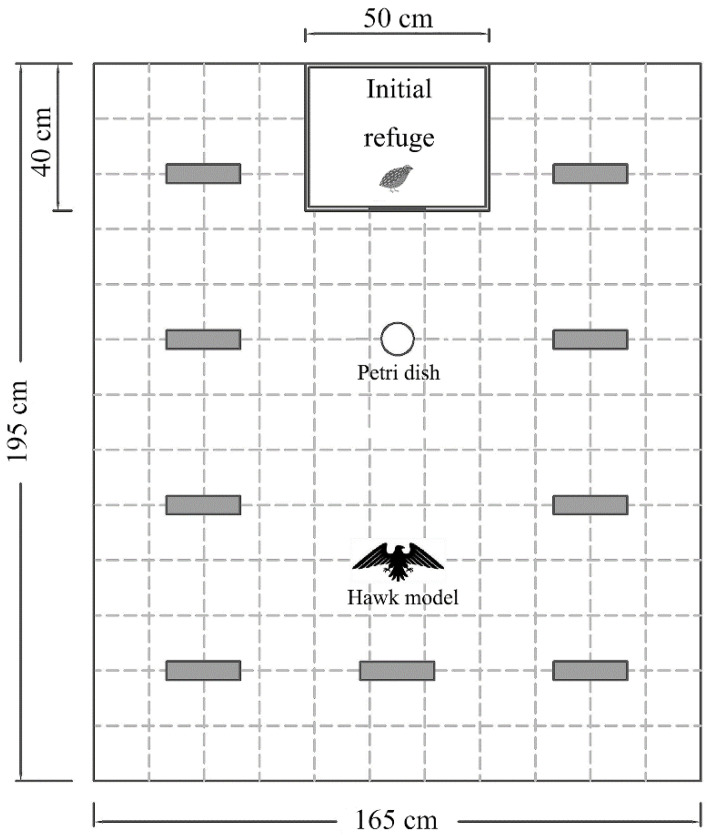
Schematic view of the experimental tank used to measure boldness and exploration.

**Figure 2 animals-11-03423-f002:**
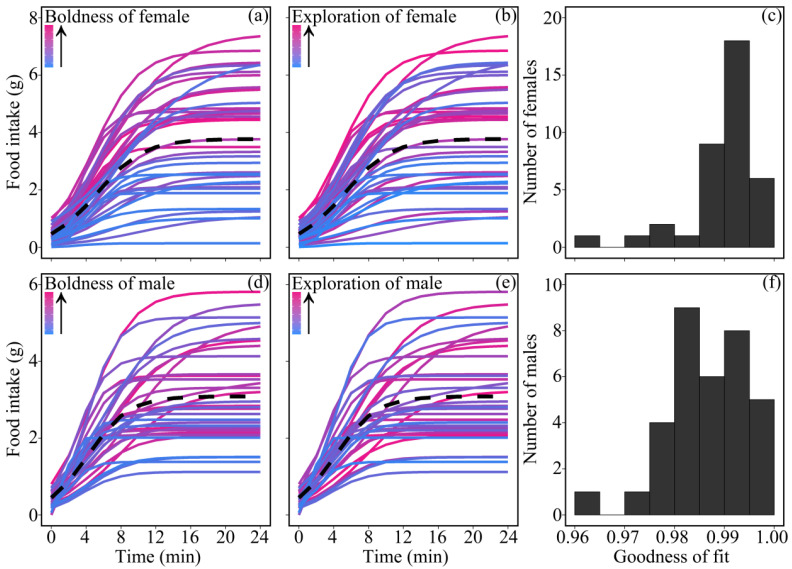
Accumulated food intake along foraging time in domestic Japanese quails. Curves in (**a**,**d**) were ordered by boldness and those in (**b**,**e**) were ordered by exploration. The dark dotted curve in each figure represents the average curve of individuals of the sex. The histograms of (**c**,**f**) display how well the accumulated food intake of females and males were fitted by the logistic model.

**Figure 3 animals-11-03423-f003:**
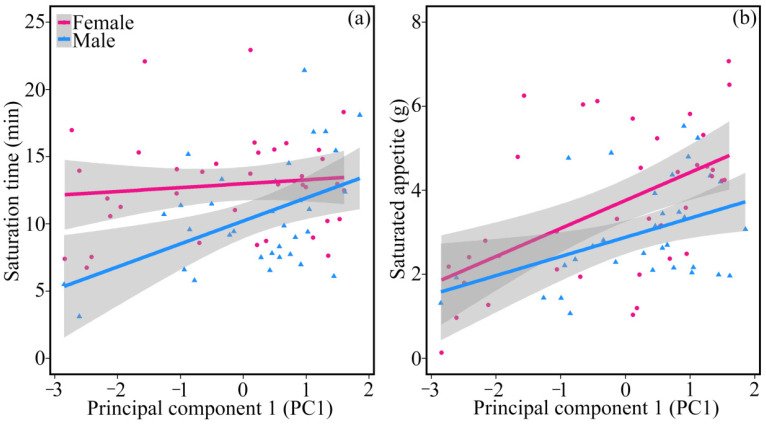
Saturation time and saturated appetite as a function of principal component 1 (PC1) of personality traits (boldness and exploration) in domestic Japanese quails. PC1 represented boldness and exploration and higher score of PC1s indicated that individuals were more proactive. (**a**) was the influence of PC1 on saturation time, (**b**) was the influence of PC1 on saturated appetite.

**Table 1 animals-11-03423-t001:** Effects of body weight, sex, and two independent principal components (PC1 and PC2) of personality traits (boldness and exploration) on four scratching behaviours in domestic Japanese quails: the total number of scratching bouts during which the quail continuously scrabbled the petri dish with its claws (NSB), the number of scrabbles during each scratching bout (NS), the time of the first scratch since the beginning of experiment (TFS), and the total time of scratching bouts (TTS).

Scratching Behaviours	Variables	Estimate	Standard Error	*t* Value	*p* Value
NSB	PC1	3.632	1.131	3.213	**0.002**
	PC2	2.068	2.421	0.854	0.396
	Body weight	−0.116	0.168	−0.686	0.495
	Sex	−1.607	3.467	−0.464	0.646
NS	PC1	10.184	4.360	2.336	**0.023**
	PC2	9.742	9.337	1.043	0.301
	Body weight	−0.471	0.649	−0.726	0.471
	Sex	2.076	13.370	0.155	0.877
TFS (s)	PC1	−259.113	37.997	−6.819	**<0.001**
	PC2	−171.719	81.374	−2.110	**0.039**
	Body weight	8.869	5.659	1.567	0.122
	Sex	170.345	116.517	1.462	0.148
TTS (s)	PC1	5.327	2.2704	2.346	**0.022**
	PC2	4.542	4.862	0.934	0.354
	Body weight	−0.295	0.338	−0.873	0.386
	Sex	−0.654	6.962	−0.094	0.926

Significant coefficients are in bold.

**Table 2 animals-11-03423-t002:** The correlation coefficients between principal component 1 (PC1) of personality traits (boldness and exploration) and food intake during each 2 min foraging period in domestic Japanese quails.

Order of 2 min Foraging Period	Pearson’s Correlation Coefficients		*p* Values	
	Females	Males	Females	Males
1	0.516	0.093	**0.003**	0.600
2	0.509	0.205	**0.003**	0.420
3	0.628	0.292	**<0.001**	0.281
4	0.468	0.438	**0.009**	**0.029**
5	0.386	0.386	**0.050**	**0.073**
6	0.383	0.388	**0.053**	**0.070**
7	0.437	0.421	**0.018**	**0.039**
8	0.267	0.388	0.211	**0.070**
9	0.263	0.477	0.222	**0.013**
10	0.222	0.476	0.360	**0.013**
11	0.136	0.515	0.828	**0.006**
12	0.193	0.454	0.493	**0.021**

Significant coefficients are in bold.

## Data Availability

The data presented in this study are available upon request from the corresponding author.

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
