# Peer review of "Effects of Personality Traits on the Food-Scratching Behaviour and Food Intake of Japanese Quail (Coturnix japonica)"

_animals, 2021, doi:10.3390/ani11123423_

Round 1

Reviewer 1 Report

Dear Authors,

Many thanks for submitting this interesting paper investigating the effect of personality on the foraging behaviour of quail. I found your paper to be well written, and the experimental design was robust. I have a couple of small queries with regards to statistical testing (see the PDF manuscript with comments) though these are all relatively minor. While personality has been well studied in a range of species, the link to foraging behaviour is useful and interesting. On the whole, I would like to commend the authors on a well-designed experiment and well developed project.

Reviewer 2 Report

This is an interesting and useful paper that provide some relevant areas of investigation into the effect of personality on important behaviours (i.e. foraging). I recommend some edits to the paper, listed below, to improve clarity and repeatability. 

Simple summary.

Replace Phasianidae with gamebirds. 

Abstract

Suggest "Overall foraging success and ultimate fitness of an individual  animal is highly dependent..."

Introduction

Suggest the first line is changed to "When individual forage optimally, they can increase their success at collecting food and minimising energetic costs... Survival is therefore dependent on performing successful foraging strategies."

Line 46, suggest you mention personality here. We know lots about how behaviour patterns impact on foraging. You are referring to individual animal personality. 

Line 77: Recommend writing as "ground-feeding birds of the Order Galliformes (the pheasants, partridge, grouse and allies) 

Line 80: the scientific name needs to be in parentheses? 

Line 81: suggest "... each quail's personality traits..."

Line 86: might there be other trade offs here? More frequent scratching may use more energy. So pro-active birds might burn more energy when looking in less profitable feeding patches. Perhaps you need to consider the "word smart, not work hard" birds and their strategy too? I.e. quail that copy the foraging strategies of others and see where they are foraging and then scratch? 

Line 92: I don't think you mean every single law ever written in China. I imagine you mean the research complied with animal welfare and scientific research ethics legislation?

Line 96: Where were the quail kept? In what facility? 

Line 113: How were the birds labelled? Legs rings? Is this how you determined which was male and which was female?

Please can you provide an ethogram that states the name of each personality trait and a description of its behavioural expression (i.e. the bird's body language and posture that would enable others to identify it). 

When you talk about "the cardboard" I think you mean the container or test arena? Might be worth replacing cardboard with arena, for example "the experimenter gently placed the Petri dish back into the arena".

Was the Petri dish put in the same area of the experimental arena for each trial with each quail?

Were all statistical analyses run in R? If so, please state this at the start of the data analysis section, including the programme version of R.

Please explain the package used for PCA.

What raw data were included in the PCA?

How was model fit determined?

You mention use of GLMs in Line 241. Were data included in this model normally distributed? Did you check the model for dispersion and check plots of residuals?

Line 248 please explain the reasoning behind the use of this formula. 

Line 256. You have calculated individual sample point correlation coefficients and then you have analysed each of these coefficients to see if they are significantly different? Are you not "over analysing" your data here and do you not run the risk of false discovery?

I think this is a very detailed statistical analysis section but I think a reader would struggle to replicate it. Are you able to explain the application of each test or model by aim or research question? And can you provide more details on how models were fitted and the fit evaluated?

Do you need the complexity of all of the different tests? 

In the results section, where you have tables of multiple P value comparison, I recommend calculating a new level of significance to check each P value against, to reduce the chances of Type 1 error. For example Benjamini-Hochberg correction. This could be useful for Table 1, for example.

I recommend using the term Galliformes in the discussion in place of Phasianidae birds.

Line 359 and 326 please provide the specific common name for the agama and the tamarin. 

Was there any effect of physiological state on foraging behaviour? Were female quail going to start laying eggs during the test period? Or would birds be entering moult? These variables will influence time spent foraging.

The discussion is quite short. Could you provide some suggestions of future research extensions that could allow you to probe more deeply into personality effects. For example, if the birds were foraging in a social group, how could that impact on their foraging activities as well as interactions with the personality of other birds?
